# Comparison of Academic Motivation between Business and Healthcare Students in Online Learning: A Concurrent Nested Mixed-Method Study

**DOI:** 10.3390/healthcare10081580

**Published:** 2022-08-19

**Authors:** Yasuhiro Kotera, Valentina Gorchakova, Sarah Maybury, Ann-Marie Edwards, Hiromasa Kotera

**Affiliations:** 1School of Health Sciences, University of Nottingham, Nottingham NG7 2TU, UK; 2College of Business, Law and Social Sciences, University of Derby, Derby DE22 1GB, UK; 3College of Health, Psychology and Social Care, University of Derby, Derby DE22 1GB, UK; 4Department Linguistics, University of Potsdam, 14476 Potsdam, Germany

**Keywords:** academic motivation, online students, intrinsic motivation, extrinsic motivation, mixed-method

## Abstract

While the demand for online education and the diversity of online students have been increasing worldwide, how online students motivate themselves to continuously engage in learning remains to be appraised. Research in the face-to-face contexts reports that academic motivation is central to student success and wellbeing, and the type of motivation can differ by subject. In particular, the motivation of business students and healthcare students can differ considerably. This study aimed to understand the motivation of online students, and compare them between business and healthcare students using a concurrent nested mixed-method design with correlation and thematic analyses. A survey regarding motivation, learning enjoyment, and study willingness was responded to by 120 online students (61 business and 59 healthcare). Business students were associated with extrinsic motivation, whereas healthcare students were associated with intrinsic motivation. While students in both groups enjoyed the pursuit of knowledge, healthcare students valued the process and accomplishment, whereas business students regarded education as steppingstones in their careers. Findings can help educators develop effective motivational support for these student groups.

## 1. Introduction

Online learning has gained significant prominence in higher education institutions (HEIs) over the past two decades, with many incorporating this into their long-term strategies [1,2]. Several advantages are associated with online learning for students, namely reduced time and geographical constraints, increased control over class pace and engagement, and fewer costs associated with them, i.e., on commuting or relocation, altogether providing a more flexible and attractive format for many [3,4]. The nature of online learning requires students to possess efficient time management skills and the ability to self-direct their learning [5]. These benefits extend to HEIs, who can reach a broader population of students and those who would otherwise be unable to access traditional face-to-face higher education [2,6].

Despite the noted unique advantages, some challenges have been reported. The reduced peer and teacher interaction can cause motivation issues [7,8,9]. In addition, as noted by Curasi and Burkhalter [10], the profile of online students is becoming increasingly untraditional, including married students with children, and students who are employed full-time. These students are less able and less motivated to dedicate additional time to course materials to complete the course, as their outside lives tend to be prioritised over their studies [10]. Additionally, a lack of belongingness to the university or a learning community sabotages their motivation [2], accompanied by a strong sense of loneliness [11]. These factors could help explain the poor performance and high attrition rates associated with online learning [12].

More specifically, online business students and healthcare students face motivational challenges. Pre-COVID studies reported that although many business students appreciated the self-paced nature of online learning (e.g., fitting studies in their work life), a lack of personal support experienced in online learning relative to face-to-face learning was challenging to business students especially those who were new to online learning [13,14]. During the COVID-19 pandemic, business studies were relatively unaffected by the forced transition to online, because many components of their curriculum were adaptable to online learning [15]. However, a lack of the social aspect of learning (e.g., discussion and conversation with lecturers and peers) was demotivating for many business students [16] Likewise, in pre-COVID research, online healthcare students reported their motivation was compromised in group work where good engagement from all group members was required [17]. To counter this problem, educators implemented measures such as mandatory attendance to online lectures and activities. However, these measures could counteract the self-paced nature of online learning, disadvantaging those with a job with irregular working hours, career duties, and living in a different time zone [18]. Unsurprisingly online students’ intrinsic motivation for asynchronous learning was higher than for synchronous learning, relating to a sense of autonomy [19]. These challenges were intensified during the COVID-19 pandemic: many students’ duties outside the curriculum have increased (e.g., deployment to hospital, obscured boundary between study and life [20], making it hard for them to stay motivated to study [21,22]. The forced transition to online learning was experienced positively by healthcare students who were undertaking desk-based learning [23], whereas those in clinical practice experienced it negatively as their practice was interrupted [24]

Understanding academic motivation continues to be of importance to educators, as a necessary function of maintaining student engagement and retention, and to encourage positive academic outcomes [25]. This has become even more pronounced in the context of the global COVID-19 pandemic, which continues to disrupt conventional teaching methods, forcing many HEIs to provide online instruction, even for those who did not originally enroll in online learning courses [5]. The demand for online learning has clearly increased worldwide.

One well-established theory of motivation is Deci and Ryan [26] Self-Determination Theory (SDT), a model that seeks to explain human needs, motivation, and wellness in various social contexts [12]. SDT argues that autonomy, competence, and relatedness are critical components of motivation [27]. It distinguishes between three types of motivation: (1) intrinsic motivation, carrying out an activity based on pleasure or enjoyment, (2) extrinsic motivation, behaviours undertaken for reasons beyond innate satisfaction, and (3) amotivation, the absence of intentionality [28]. Within the context of education, the focus of SDT is on promoting student self-confidence, the value of education, and a desire to learn [29]. Both intrinsic and extrinsic motivation act as a mediating role for academic performance through online learning behaviours [30]. Those are essential components of education [29], indicating the relevance and importance of motivation in education.

The findings of Griffin et al. suggest that the level of intrinsic motivation demonstrated by students constitutes the most powerful indicator of student success. Students with a high level of intrinsic motivation tend to exhibit increased engagement in classes, experience positive self-esteem and a sense of subjective wellbeing, and excel academically [31]. In contrast, the predominance of extrinsic motivation can result in negative outcomes such as boredom, poor class attendance, and academic performance, and amotivation is negatively associated with engagement, learning potential, and wellbeing [28,32]. This is important as many students have adopted a consumerist approach to higher education, where extrinsic motivators (i.e., grades) take precedence, fostering an environment of shallow cognitive engagement [29,33]. Not only can motivation be a powerful predictor of academic achievement, but it also can impact decisions to commence and continue specific learning behaviours, helping students become successful professionals [34]. Moreover, intrinsic motivation is a known predictor of student psychological well-being, a significant determinant of the decision to remain enrolled in their course [35,36]. The positive impacts of intrinsic motivation are diverse.

Educators and researchers have begun to pay attention to interventions to support students’ intrinsic motivation. However, this has encountered several challenges due to educational goals and motivation types that differ across academic disciplines [37,38]. Those disciplines that are perceived as ‘hard’, such as business, management, and technology, place more emphasis on the application of what is learned versus the knowledge itself, attracting students with high levels of extraversion, linked to high extrinsic motivation [37,39]. For example, although some shift towards intrinsic motivators such as meaning is reported among business students [40], a business degree is highly sought due to the reputation of high earnings potential and elevated social image [39,41]. Additionally, it is thought that intrinsic motivation is stimulated in some contexts in business education [42]. Cheng et al. [43] highlighted that in the context of business students pursuing personal certifications, the prevalent use of rewards (i.e., monetary) acts as reinforcers of extrinsic motivation and overshadows the potential for intrinsic motivators, such as a sense of belonging and ability. Furthermore, in an investigation of motivation in business students in India undertaking an MBA, the brand name of an institution was a main motivator based on its relevance to career prospects [44]. These studies share a common aspect whereby the pursuit of a business degree is perceived as a necessary step to reach the desired goal of a career and life in the corporate world, rather than the standalone merits that a business education may provide.

On the other hand, while some extrinsic rewards do act as an attractor [45], many healthcare students, categorised as ‘soft’ disciplines, are inspired by intrinsic motivators such as an altruistic desire to help others [46,47]. For example, nursing is often seen as a ‘calling’ rather than a job, where the opportunity to care for others, being part of a team, and a sense of pride in the services provided are cited as primary drivers of pursuing a nursing degree and subsequent career [48,49]. When examining motivation in students taking nutrition classes, Maurer et al. [29] found these students to be principally driven by intrinsic motivation, specifically the contribution that the course would have to their understanding of their own health status. In a study examining career choice and motivation of medical students and the potential impact on their academic performance, almost 75% of participants cited reasons that related to intrinsic motivation [35]. These reasons predominantly included intellectual curiosity, and experience of illness or death either personally or within their family Kim et al. [35], highlighting the relevance of intrinsic motivation in healthcare students.

These intrinsic drivers may become threatened when more external rewards are used [50]. External rewards such as money ‘crowd out’ intrinsic motivation, shifting the emphasis from engaging in a task for internal feelings of self-determination, to obtaining the reward itself [51,52]. Furthermore, while intrinsic motivation and extrinsic motivation are not dichotomous (i.e., one individual can have high levels of both intrinsic and extrinsic motivation), often a larger external reward could spoil intrinsic motivation, leaving extrinsic motivation as the dominant driver of behaviour [53]. The implications of the predominance of extrinsic motivation in students extend beyond academic achievement and success. In a study investigating approaches to recruiting peer mentors, Anghelcev and Eighmey [54] found that students receiving no monetary reward were the most willing to become peer mentors, followed by those receiving a low reward, and those receiving the highest monetary value were the least willing to participate. Moreover, the use of external rewards such as a desire for recognition has implications for enjoyment. Children who are attached to parents’ praises may lose a sense of enjoyment in an originally enjoyed behaviour [55], whereas nurturing a sense of competence and autonomy (components of intrinsic motivation) in students can lead to elevated feelings of enjoyment towards a certain task and may promote positive cognitive and behavioural outcomes [56]. Despite these findings, researchers remain divided over the impact of external rewards [57]. Therefore, it is important to evaluate whether a) students enjoy studying the subject (enjoyment), and b) students would still study in the programme if they were not to receive a grade (willingness), and how those are related to each type of motivation.

Considering the significance of furthering our understanding of motivation in learning, it is crucial to explore the academic motivations of students across different academic disciplines, particularly in the context of online learning. Accordingly, this study sought to add to the literature on the similarities and differences between the types of motivation among healthcare and business online students. Four research questions were established.

RQ1: Are the levels of each type of motivation different between business and healthcare students?

RQ2: How is each type of motivation related to each other?

RQ3: How is each type of motivation related to learning enjoyment and willingness to study without any external reward?

RQ4: What are the similarities and differences in key academic motivations and goals of business and healthcare students?

## 2. Materials and Methods

### 2.1. Study Design

This study employed a concurrent nested mixed-method design [58] Qualitative findings (secondary focus) were used to complement the quantitative findings (primary focus), to gain a better understanding of the students’ motivation.

### 2.2. Study Context

The study was conducted at an online department of a university in the United Kingdom, where almost all students are working professionals. Undergraduate business and healthcare students were recruited from January to June 2021 using an online survey link embedded in the programme announcements delivered by an academic who was not a co-author of this article. The survey included quantitative and qualitative questions. About 300 students were approached in each discipline.

### 2.3. Participants and Recruitment

One hundred and twenty students completed the survey in total. Sixty-one business students (36 females and 25 males; Age M = 33.59, SD = 10.56, Range 18–57 years) consisted of 31 UK students and 30 international students (11 other Europeans, 6 Asians, 5 Africans, and 1 Oceanian: 7 did not report). All business students were enrolled in a business and management programme. Fifty-nine healthcare students (48 females and 11 males; Age M = 45.28, SD = 11.24, Range 25–70 years) consisted of 38 UK students and 21 international students (7 other Europeans, 4 Africans, 2 Asians, and 1 Oceanian: 7 did not report). Thirty-six students were in counselling, 17 in nursing, 4 in social care and 2 in perioperative practice programmes. As reliable data were not available, the representativeness of our samples with the general populations of business and healthcare students was not certain, however, our samples’ demographics were roughly comparable with the entire cohorts [59,60]. Demographic data were summarised in Table 1.

### 2.4. Data Collection

The research ethics committee at the university approved this study. Consent was received from all participants before participation. Participants were informed at the beginning of the study that (a) the participation was arbitrary, (b) they were allowed to withdraw at any time just by closing the browser, (c) reasons for withdrawal would not be asked, and (d) all responses would be anonymised, and (e) the study might be published or presented for academic purposes. Should a participant feel distressed while completing the questionnaire, contact details of the student wellbeing services in the university were introduced.

### 2.5. Instruments

#### 2.5.1. Quantitative

The quantitative component of the survey consisted of the Academic Motivation Scale (AMS; [61]) and two additional yes/no questions.

AMS is a 28-item scale that evaluates the levels of three types of motivation, which are further categorised into seven subtypes measured by four items each: (A) amotivation, (B) extrinsic motivation (external, introjected, and identified regulation), and (C) intrinsic motivation (to know, to accomplish, and to experience stimulation). Example items include ‘I can’t see why I go to college and frankly, I couldn’t care less’ for amotivation, ‘In order to have a better salary later on’ for extrinsic motivation, and ‘For the pleasure I experience when I discover new things never seen before’ for intrinsic motivation, which are responded on a seven-point Likert scale (from 1 = ‘Does not correspond at all’ to 7 = ‘Corresponds exactly’). All of the subscales have adequate to high reliability with Cronbach’s alphas between 0.62 and 0.91 [61].

Based on the assumptions of SDT, the Academic Motivation Scale (AMS) was formulated by Vallerand et al. (1992) [61] as an instrument to measure intrinsic motivation, extrinsic motivation, and amotivation in students. While the AMS has been extensively researched to investigate motivation in high school and college generally, more work can be conducted to apply the AMS to specific disciplines [29]. Therefore, the AMS was used in the current study to assess motivation levels among participants.

The two yes/no questions were ‘Do you enjoy the subject by itself?’ and ‘If no rewards are given (just pure joy of learning experience), would you still study?’. In the analysis, ‘yes’ was coded as ‘1’, and ‘no’ as ‘0’.

#### 2.5.2. Qualitative

The seven items in the qualitative component were (1) ‘Why did you decide to study online?’, (2) ‘What motivates you to engage in academic activities?’, (3) ‘What is/are your main goal(s) of study?’, (4) ‘What do you want to achieve in this online programme?’. (5) ‘What motivates you to engage in learning?’, (6) ‘What hinders you from engaging in learning?’, and (7) ‘Is there anything else that you feel I should have asked, or that you would like to add?’.

### 2.6. Analyses

To respond to RQ1–3, a correlation analysis was conducted. The point biserial correlation coefficient was used for the students’ discipline (0 = Healthcare, 1 = Business) to address RQ1. For the qualitative data (RQ4), thematic analysis was conducted to capture the themes in the participants’ text responses. Following Braun and Clarke’s guidelines [62,63], the text responses were read several times, and codes were generated (Table 2), which led to the creation of themes (Table 3). Themes were reviewed by all co-authors and reached an agreement.

## 3. Results

### 3.1. Quantitative Analysis

No outliers were identified. P-p plots demonstrated that the quantitative data were normally distributed. The levels of each type of motivation were summarised in Table 4. The reliability of each motivation type was high in both student groups (α = 0.75–0.88 in business, α = 0.72–0.84 in healthcare). Pearson correlation was performed (Table 5).

Students’ discipline was negatively associated with intrinsic motivation to know (IMK), intrinsic motivation to experience stimulation (IMS) and no reward, while positively associated with extrinsic motivation for external regulation (EMER), suggesting that healthcare students had significantly higher levels of IMK, IMS and willingness to study without reward, and lower EMER than business students (RQ1). Overall, intrinsic motivation and extrinsic motivation were positively interrelated, whereas extrinsic motivation and amotivation were not (RQ2). Lastly, only IMK was associated with learning enjoyment among all types of motivation. All three types of intrinsic motivation (to know, to accomplish, and to experience stimulation) were positively associated, and EMER was negatively associated with willingness to study without reward (RQ3).

### 3.2. Qualitative Analysis

Thematic analysis on responses of business and healthcare students has revealed three prominent themes, based on one similarity and two differences in the motivating drivers to engage in academic studies and in learning in general (Table 2). The motivations shared by students of both disciplines included the following.

Striving for the Pursuit of Knowledge (T1)

For many healthcare students, the opportunity to learn more and enhance their understanding of certain subjects, i.e., intrinsic rewards, was both a major motivating factor in engaging with course tasks, and their desired outcome: The prospect of obtaining a reward was perceived as secondary.


*HCS (Healthcare Student): I have not been rewarded for my learning because I do this for myself. I would still study without rewards, after all studying itself is rewarding.*


This was also the case for quite a few business students, one expressing with certainty their willingness to study without the prospect of reward or recognition.


*BS (Business Student): Absolutely. Learning is a gift. Knowledge is power and the one thing that no one can take away. There is pride and joy in learning.*


One healthcare student expressed a degree of opposition towards the concept of receiving rewards for their studies, viewing this as a hindrance to the quality of their learning experience.


*HCS: In some ways, I find it distracting to have to study with a specific agenda in mind, rather than just being able to explore the materials at my leisure and to discover for myself the parts that are relevant and interesting to me.*


These comments suggest that strong orientation towards intrinsic motivation in healthcare students. On the other hand, business students, who were employed, were often enrolled in the course solely on the request of their employers who were providing the funds, and therefore had conditions attached to them. Moreover, some business students noted that consideration for certain jobs and promotions were contingent on obtaining specific degrees, which can lead to high extrinsic motivation.


*BS: My employer is paying my fees which motivates me.*

*BS: I did it as a part-sponsored degree within my work to help me achieve a promotion and better prospects for the future.*


Contrary to healthcare students, these comments by business students indicate a strong connection between their degree and employment. That said, there were some responses from healthcare students which demonstrated clear extrinsic motivation, whether to continue their professional development, or to work towards a successful career.


*HCS: To supplement my existing undergraduate, post-graduate, and master’s level qualifications with a further accreditation relevant to my chosen profession.*

*HCS: Future career development, to be more competent, to be internationally recognised, for more salary, etc.*


These responses indicate that many business- and healthcare-related degrees are structured for the purpose of employment, and to fit the needs of employers. When combined with the costs of obtaining a degree, the prevalence of extrinsic motivation in participant responses can somewhat be expected.

Responses provided by both business and healthcare students also suggest the fluidity of motivation: intrinsic and extrinsic motivation can change. While some students enrolled in their course specifically for practical (extrinsic) considerations, they noted that over time they began to value the course for the content alone, and the opportunity to learn.


*HCS: It was initially to get promotion as BSc Nursing was needed to be promoted to level 13, in the place I work some have been in level 12 for 11–12 years... But now I found academics very interesting though can be difficult and challenging.*


Moreover, another participant highlighted a contextual impact on motivation. Their motivation was shaped by their current circumstances and responsibilities, noting that once they were in a better position, they would study for the passion of learning itself.


*BS: I do enjoy learning new things, but at the moment my focus is on obtaining the end reward of a bachelor’s degree. However, in the future, once I am in a comfortable working position and have achieved what I wanted, I would continue learning for the learning experience.*


Learning Process and Accomplishment in Healthcare Students (T2)

Many healthcare students noted that the process of learning and a sense of accomplishment as key motivators. Not only is the knowledge gained from a university education a main motivator for students, but the different steps that comprise the learning process was another major theme for healthcare students.


*HCS: I am not only exercising my physical muscles but also my mental and cognitive abilities give me an immeasurable amount of satisfaction.*

*HCS: I feel a sense of achievement in the process [of] taking part and pushing my comfort zones.*


These comments suggest that the opportunity to be in an environment where they can challenge themselves while developing their mental acuity is their major motivator. The learning process is personal, acting as a boost to their self-esteem, rather than an instrument to prove their value over others.


*HCS: It feels really good to learn something that is useful, it doesn’t cause me to feel more important than people who don’t have qualifications: it is more I am fulfilling my own potential no matter how old I am.*


The majority of online healthcare students at our institution are registered professionals in employment, and many of them have other non-academic commitments (e.g., childcaring). Their life context is often more restricted than typical young students who have just graduated from high school to afford an education. Therefore, online healthcare students reported a sense of accomplishment to manage their studies in online learning.

Their degree provides them with a sense of satisfaction. Their motivation to engage with their course and desired outcome were intrinsically oriented.


*HCS: Feelings of validation in my abilities [are my motivation to study].*

*HCS: To fulfill my potential which did not happen while I was at school. To be a “good” role model to my two daughters.*


Taken together, many healthcare students enjoy the process of learning to expand their knowledge and skills, as well as a sense of accomplishment derived from progressing their studies (i.e., intrinsic motivators) while managing other duties.

Education as Steppingstone for Business Students (T3)

Many business students perceived university education as a steppingstone to advancing their careers. While they acknowledged the importance of the learning process, they also noted the social value attached to getting a university education.

A university education was treated as an element of social capital, providing a sense of membership and common understanding based on shared experience. Being a university graduate comes with baseline assumptions in relation to intelligence level, organisational and time management skills, interpersonal skills, and to some degree socioeconomic background. Therefore, those who do not fall under the category of university graduate, may be made to feel like an outsider.


*BS: [I] felt inferior [to] all of my friends and colleagues [who] are university graduates.*

*BS: My full degree to enable me to apply for certain roles that require minimum 2:1.*

*BS: To have a job where I can earn a good salary.*

*BS: Get the necessary UK qualification to enter/access a new work sector.*


Online business students, who already worked in business at the time of the study, expressed feelings of inadequacy because, unlike their colleagues, they do not have a university degree. Therefore, the motivation at the forefront for them appears to be utilising their degree as a resource to achieve a level playing field with their social circle. Similarly, other business students noted a desire to demonstrate their ability by obtaining a degree. In response to the question around the primary motivation for enrolling in the course and engaging in their unit activities, two business students provided the following answers.


*BS: To achieve higher grades and show my work to my peers.*

*BS: Hmm… like the letters after my name to be honest.*


These comments suggest that business students regard university education as a steppingstone to enhance their credibility, and being more accepted in their business community.

### 3.3. Synthesis

The findings in our quantitative analyses and qualitative analyses were roughly consistent with each other: healthcare students were, in general, more intrinsically oriented than business students, and intrinsic motivation was associated with learning enjoyment. Our qualitative analyses deepened these findings that both groups were motivated to pursue knowledge, however healthcare students enjoyed the learning process whereas business students somewhat regard education as a steppingstone in their career.

## 4. Discussion

The purpose of this study was to evaluate the academic motivation of online business and healthcare students. Our results show that (RQ1) healthcare students had a higher level of intrinsic motivation, whereas business students had a higher level of extrinsic motivation, (RQ2) intrinsic motivation and extrinsic motivation were positively inter-related, (RQ3) intrinsic motivation was associated with both enjoyment and intention to study, and (RQ4) while students in both groups enjoyed the pursuit of knowledge, healthcare students valued the learning process and accomplishment, whereas business students regarded education as a steppingstone.

One novel finding of our mixed-method analyses may be that although the demographic data were different between online students and face-to-face students (e.g., age and life stage), their motivational characteristics were similar. Previous face-to-face research reported a high level of extrinsic motivation in business students, and a high level of intrinsic motivation in healthcare students [36,64,65,66]. We found the same motivation characteristics were present in online students. Moreover, we further identified their perceptions towards learning in relation to motivation: healthcare students were process-oriented, whereas business students were more focused on their career steps. These findings have several implications. For example, to help maintain academic motivation, educators can focus on the learning process in healthcare programmes whereas educators in business can link their curriculum to student career steps.

High intrinsic motivation was associated with healthcare students, whereas high extrinsic motivation was associated with business students. Healthcare students enjoyed the learning process and accomplishment, both of which are rather intrinsic rewards. On the other hand, business students consider education as a steppingstone, an extrinsic reward. In line with the face-to-face research [39], academic motivation was different between these two groups in the online context: in the face-to-face setting, business students were more associated with extrinsic motivation, whereas healthcare students were more associated with intrinsic motivation as well. Moreover, this trend is present when these students have become professionals/employees [52,67,68,69]. These findings suggest that securing a prestigious and highly salaried position is the primary driver for choosing a business education programme [70], whereas healthcare students are more likely to study a healthcare subject out of personal interest or enjoyment [35], leading to improved learning outcomes. These characteristics were deemed similar (a) between online and face-to-face education, and (b) between pre-graduation and post-graduation. Indeed, many healthcare students see working in the healthcare setting as a vocation rather than a job, with a sense of calling to a particular healthcare profession, the basis of patient-centred and compassionate care [71]. A study conducted by Wu et al. (2020) [72] found that intrinsic motivation was significantly and positively associated with self-efficacy, learning engagement, and academic performance in medical students. This suggests that engaging with student/employee motivation should be conducted over time, including when they are preparing to work, and when they are working.

Considering these positive associations of intrinsic motivation, business students could benefit from shifting their extrinsic motivation to an intrinsic one. Cognitive evaluation theory informs that incorporating engaging learning activities such as reading stimulating materials and self-awareness diaries helps students focus on their grades to their learning [73]. Moreover, enhancing self-compassion may be helpful too as self-compassion assists the extrinsic to intrinsic shift [52,73]. These are consistent with the significant inter-relations between intrinsic and extrinsic motivation. Business educators need to embed this training and education in their curriculum to prevent business students from experiencing the negative consequences of extrinsic motivation. It is also essential to mention that intrinsic motivation can be harmful to students if their passion is obsessive (based on uncontrollable compulsion) instead of harmonious (balanced engagement; [65,74]). Self-awareness needs to be encouraged among healthcare students [75,76]. Moreover, during the pandemic, a sense of loneliness was identified as a debilitating factor for healthcare students’ well-being and motivation [77]. To improve motivation (i.e., to cope with amotivation) psycho-social support was recommended to augment a sense of connection and belongingness for healthcare students [78,79]. Post-traumatic growth, which many healthcare students are related to, due to traumatic events in their practice, is associated with intrinsic motivation, and career support was recommended to support their intrinsic motivation [80,81,82]. In sum, both extrinsic motivation and intrinsic motivation can contribute to academic performance, and these types of motivation can change across different phases of the curriculum influenced by various intrapersonal and interpersonal factors [83,84]. Further understanding of motivation among university staff is recommended.

Significant associations between intrinsic motivation and enjoyment, and willingness (and no significant associations between extrinsic motivation and enjoyment, and willingness) were consistent with the SDT that students who are passionate about the subject tend to enjoy studying and be more willing to study without external rewards. Especially, the willingness to study was significantly associated with healthcare students, which can be further explained using the qualitative findings.

Our qualitative analyses helped to understand our quantitative findings more holistically. Both student groups enjoyed learning, which is related to the intrinsic motivation to maintain the desire to continue the learning process [85]. Students who are intrinsically motivated are self-driven, challenged, and find enjoyment in their studies [86]. Student involvement in the learning process increases when students find study engaging, effective, and valuable [87]. Online professional students who chose to engage in a further degree have intrinsic motivation to enjoy the pursuit of knowledge.

The different perspectives on learning between healthcare and business students deepened our quantitative findings: healthcare students focused on the process and a sense of accomplishment from learning, whereas business students considered their career ahead and regarded education as a steppingstone. Generally speaking, the process of and accomplishment in learning are pertinent to intrinsic motivation, whereas a focus on further career steps is more pertinent to extrinsic motivation [28]. However, healthcare educators may need to be mindful that, as noted above, an obsessive need for accomplishment can trigger extrinsic motivation. If the standard for accomplishment is set extrinsically, social comparison can take place, leading to a more negative evaluation of oneself [88]. Self-awareness training may be helpful for healthcare students to remain intrinsic towards their learning [89]. On the other hand, for business students, how to bring their external focus inward would be more important. Interventions such as life crafting or ikigai (a Japanese concept roughly translated as a meaning in life) approaches may be helpful as these ask students about the intrinsic meaning of their career goals [90,91]. Future research needs to evaluate the impact of those approaches on students’ motivation and wellbeing.

While this research provides valuable insight into the motivation of business and healthcare students, it is important to keep the limitations in mind when interpreting the results. First, in addition to the modest size, our sample was recruited through convenience sampling from one academic institution in the UK, limiting our findings’ generalisability. Moreover, considering the wide impact of COVID-19 on academic motivation, students in other subjects should be evaluated. Secondly, a self-report survey was used, thus response biases might have been present [92]. Additionally, a shorter version of the Academic Motivation Scale [93] could have been used to reduce the workload of participants, improving the accuracy and response rate. Moreover, in the qualitative part, participants’ emotional tone was not captured in this study. Other possible motivational factors such as the employment status [94] were not considered in this study. Relatedly, the impact of COVID-19 was not assessed. These two factors could have a meaningful impact on our findings. For example, an unstable employment status negatively affected people’s psycho-social well-being [95]. During the pandemic, university students’ academic motivation was compromised, as their sense of belonging declined [96]. Future research needs to consider these factors. Lastly, the study does not examine whether students had specifically chosen to study online or whether e-learning was the only option available to them [97]. Future research needs to address these limitations to identify more robust evidence for online students’ motivation.

## 5. Conclusions

Academic motivation is essential for student success, however, it remains to be appraised in online education, which has been gaining importance today. In particular, business and healthcare students learning online encounter some motivational challenges. Our study identified that business students were extrinsically motivated to regard education as a steppingstone, whereas healthcare students were intrinsically motivated to focus on the process and accomplishment. Self-compassion and meaning-oriented interventions may be helpful for business students, and self-awareness training, psycho-social support and career guidance may be useful for healthcare students to support their wellbeing. Our findings will help educators and well-being services develop effectively, and tailor mental health care in those student groups.

## Figures and Tables

**Table 1 healthcare-10-01580-t001:** Demographic data of participants.

Demographics	Business Students (*n* = 61)	Healthcare Students (*n* = 59)
**Gender: *n* (%)**		
Female	36 (59%)	48 (81%)
Male	25 (41%)	11 (19%)
**Age**		
Mean	33.59	45.28
SD	10.56	11.24
Range	18–57	25–70
**Residence: *n* (%)**		
UK (Home)	31 (51%)	38 (64%)
Other Europe	11 (18%)	7 (12%)
Asian	6 (10%)	2 (3%)
African	5 (8%)	4 (7%)
Oceanian	1 (2%)	1 (2%)
Unanswered	7 (11%)	7 (12%)
**Programme: *n* (%)**		
Business and Management	61 (100%)	
Counselling		36 (61%)
Nursing		17 (29%)
Social Care		4 (7%)
Perioperative		2 (3%)

**Table 2 healthcare-10-01580-t002:** Generating initial codes (examples).

Focus Area	Initial Codes
Healthcare Students	Self-improvement
	Pleasure of learning
	Challenging oneself
	Intellectual curiosity
	Staying current
	Sense of discovery
	Empowerment
	Adapting to change
	Self-reflection
	Professional development
	Unlimited accomplishment
	Deeper understanding
Business Students	Increased success
	International recognition
	Perceived competence
	Screening for employers
	Enhanced credibility
	Pay rises
	Sense of satisfaction
	Contingency plan
	Professional qualifications
	Foundation for career
	Potential promotion
	Social capital

**Table 3 healthcare-10-01580-t003:** Themes and examples of participant comments.

Themes	Example of Participant Comment
T1: Striving for the Pursuit of Knowledge	I have not been rewarded for my learning because I do this for myself. I would still study without rewards, after all studying itself is rewarding (HCS). Absolutely. Learning is a gift. Knowledge is power and the one thing that no one can take away. There is pride and joy in learning (BS).
T2: Learning Process and Second Chance in Healthcare Students	It feels really good to learn something that is useful, it doesn`t cause me to feel more important than people who don’t have qualifications: it is more I am fulfilling my own potential no matter how old I am (HCS).
T3: Education as Steppingstone for Business Students	I felt inferior to all of my friends and colleagues who are university graduates (BS).

HCS = Healthcare Students; BS = Business Students.

**Table 4 healthcare-10-01580-t004:** Levels of each type of motivation in business students (n = 61) and healthcare students (n = 59).

	Business Students			Healthcare Students		
	M	SD	α	M	SD	α
IMK	5.45	1.17	0.88	5.99	0.94	0.79
IMA	5.11	1.46	0.87	5.29	1.18	0.75
IMS	3.73	1.40	0.86	4.42	1.28	0.72
EMID	5.16	1.33	0.75	5.10	1.45	0.75
EMINT	4.72	1.63	0.85	4.50	1.73	0.82
EMER	4.73	1.73	0.86	4.00	1.74	0.84
AM	1.39	.86	0.82	1.31	0.73	0.78

IMK = Intrinsic motivation to know. IMA = Intrinsic motivation to accomplish. IMS = Intrinsic motivation to experience stimulation. EMID = Extrinsic motivation for identified regulation. EMINT = Extrinsic motivation for introjected regulation. EMER = Extrinsic motivation for external regulation. AM = Amotivation.

**Table 5 healthcare-10-01580-t005:** Correlation between the discipline, gender, age, nationality, types of motivation, learning enjoyment, and willingness to study without reward among online students (n = 120).

		1	2	3	4	5	6	7	8	9	10	11	12	13
1	Discipline (0 = HC, 1 = Biz)	−												
2	GN (0 = F, 1 = M)	0.23 *	−											
3	Age	−0.46 **	−0.15	−										
4	Nationality (0 = UK, 1 = Overseas)	0.14	0.24 **	−0.17	−									
5	IMK	−0.24 **	−0.13	0.24 **	−0.13	−								
6	IMA	−0.07	−0.21 *	0.19 *	−0.22 *	0.67 **	−							
7	IMS	−0.25 **	−0.10	0.29 **	−0.02	0.61 **	0.61 **	−						
8	EMID	0.04	0.05	−0.23 *	0.11	0.21 *	0.18 *	0.20 *	−					
9	EMINT	0.033	−0.10	−0.04	−0.23 *	0.22 *	0.56 **	0.36 **	0.30 **	−				
10	EMER	0.19 *	0.03	−0.31 **	0.15	−0.02	0.19 *	0.12	0.76 **	0.44 **	−			
11	AM	0.05	0.27 **	−0.14	0.12	−0.29 **	−0.29 **	−0.02	0.02	−0.07	0.08	−		
12	Enjoy (0 = No, 1 = Yes)	−0.07	0.05	0.06	0.01	0.24 *	0.10	0.07	0.03	0.03	−0.07	−0.06	−	
13	Willingness (0 = No, 1 = Yes)	−0.30 **	−0.15	0.12	0.07	0.38 **	0.20 *	0.23 *	−0.14	−0.13	−0.23 *	−0.09	0.20 *	−

HC = Healthcare. Biz = Business. IMK = Intrinsic motivation to know. IMA = Intrinsic motivation to accomplish. IMS = Intrinsic motivation to experience stimulation. EMID = Extrinsic motivation for identified regulation. EMINT = Extrinsic motivation for introjected regulation. EMER = Extrinsic motivation for external regulation. AM = Amotivation. Enjoy = Whether they enjoy learning the subject. Willingness = Whether they would be still willing to study the subject if there were no reward given for completing the programme. * *p* < 0.05, ** *p* < 0.01.

## Data Availability

The data that support the findings of this study are available on request from the corresponding author. The data are not publicly available due to privacy or ethical restrictions.

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
