# Peer review of "Comparison of Academic Motivation between Business and Healthcare Students in Online Learning: A Concurrent Nested Mixed-Method Study"

_healthcare, 2022, doi:10.3390/healthcare10081580_

Round 1

Reviewer 1 Report

This is an interesting mixed-methods study on student motivation in online courses in health and business fields.

Line 90-is an incomplete sentence and needs completion

Line 50- citation needs to be added for concurrent nested mixed-methods design. Also, why is ' nested' mentioned in this design?

Discussion is well written though needs more strengthening. Did the authors compare the motivation of online students towards face to face courses from past studies?  What does the previous  research mention about student motivation in online learning in fields other than health care and business? Are the findings comparable or different? Did the authors find out if student motivation differed based on asynchronous or synchronous online courses?

Author Response

Response Letter

Manuscript ID: healthcare-1858411

"Comparison of academic motivation between business and healthcare students in online learning: A concurrent nested mixed-method study”

Dear Reviewers,

Thank you for your helpful feedback. We have systematically revised our manuscript addressing the points you have raised. Please see our responses below. We hope this revised paper is now acceptable for publication. We extend our sincere gratitude to you for your feedback that has significantly helped to strengthen the paper.

Reviewer 1

Reviewer 1’s comment 1

This is an interesting mixed-methods study on student motivation in online courses in health and business fields.

Line 90-is an incomplete sentence and needs completion

Authors’ response 1-1

Thank you for your helpful feedback. Now the sentence is complete.

Reviewer 1’s comment 2

Line 150- citation needs to be added for concurrent nested mixed-methods design. Also, why is 'nested' mentioned in this design?

Authors’ response 1-2

In line with your comment, a reference is added. ‘Nested’ specifies a type of concurrent designs such as concurrent triangulation design or concurrent transformative design, describing that our study had one primary focus supplemented by a secondary focus. We believe the addition of the reference, as you suggested, will address this too.

Reviewer 1’s comment 3

Discussion is well written though needs more strengthening. Did the authors compare the motivation of online students towards face to face courses from past studies?  What does the previous  research mention about student motivation in online learning in fields other than health care and business? Are the findings comparable or different? Did the authors find out if student motivation differed based on asynchronous or synchronous online courses?

Authors’ response 1-3

In line with your helpful comment, academic motivation in face-to-face education, and the difference between asynchronous and synchronous online courses are discussed. We did not explore online students in other subjects. This is now added to the limitation.

Reviewer 2 Report

The aim of this manuscript is to evaluate the academic motivation of online students, focusing on business and healthcare students. It employed a concurrent nested mixed-method design, that gave priority to quantitative data, while qualitative findings were used to gain a better understanding of the students’ motivation (intrinsic/extrinsic). The quantitative analysis was performed by the Academic Motivation Scale, the qualitative module included seven items. The results showed that high intrinsic motivation was associated with healthcare students, while high extrinsic motivation was associated with business students.

The research design is appropriate, and quantitative and qualitative aspects of the study are clearly linked.

As regards the topic, this manuscript touches a significant area, and may represent a considerable contribute to the ongoing educational research since there is few evidence on this topic.

 Even if the manuscript provides an organic outline, I have some points to mention, which may help to improve its quality. For these reasons, the manuscript requires major changes.

 Please find below a list of comments on my review of the manuscript:

Introduction

 Major points

 LINES 32-49

As the main object of the study includes health and business students, rather than a general analysis of online teaching advantages and disadvantages, a more detailed explanation of challenges in these specific groups should be provided. 

Business schools are in a particularly advantageous position to adopt online teaching because business studies do not require as much practical skills as medical studies (see Szopiński T, Bachnik K. Student evaluation of online learning during the COVID-19 pandemic. Technol Forecast Soc Change. 2022;174:121203. doi:10.1016/j.techfore.2021.121203).

In healthcare schools, online teaching is suitable for pre-clinical courses (see Bianchi S., Gatto R., Fabiani L. Effects of the SARS-CoV-2 pandemic on medical education in Italy: Considerations and tips. EuroMediterranean Biomed. J. 2020;15:100–101. doi: 10.3269/1970-5492.2020.15.24), however, clinical courses were negatively affected by the interruption of internships (see Vijayan, R. Teaching and Learning during the COVID-19 Pandemic: A Topic Modeling Study. Educ. Sci. 2021, 11, 347. https://doi.org/10.3390/educsci11070347).

 Expanding this section would ensure a fair balance between the advantages and disadvantages of the two groups, leading the reader from a general subject area to a particular field.

LINE 67 AND FOLLOWING

A reference about the role of intrinsic/extrinsic motivation on academic performance in online settings is required (see Meng, X. and Hu, Z. 2022, "The relationship between student motivation and academic performance: the mediating role of online learning behavior", Quality Assurance in Education. https://doi.org/10.1108/QAE-02-2022-0046)

 Minor points

 Line 61 and following

Abbreviations should be defined the first time they appear in the main text and should be added in parentheses after the written-out form (intrinsic/extrinsic motivation). Please use abbreviations throughout the text.

Materials and Methods

Major points

 Please add a table with demographic data of the participants including the employment status

Minor points

 Line 183 the paragraph refers only to the quantitative analysis so, to avoid confusing the reader, please delete the sentence “and a qualitative component comprising seven questions.”

Line 204

I think there is a mistake in the heading, probably you meant “qualitative”

Results

Major points

 The employment situation affects students' motivation, so the following concepts should be properly in included in materials and methods (in the table of demographic data) and addressed and clarified in the discussion section:

 Lines 277-278 “business students were often enrolled in the course solely on the request of their employers”

 Lines 327-330 “The majority of online healthcare students at our institution are registered professionals, and many of them have other non-academic commitments (e.g., childcaring). Their life context is often more restricted than typical young students who have just graduated from high school to afford education.”

Discussion

 Major points

 Please provide a summary of both qualitative and quantitative results at the beginning of the section and then start the discussion in a comprehensive way. The separation of the two section weighs down the reading and generates repetitions of sentences and concepts.

 Since the study was conducted during pandemic a brief analysis of the pandemic impact on students' academic motivation, is required (See Marler, E. K., Bruce, M. J., Abaoud, A., Henrichsen, C., Suksatan, W., Homvisetvongsa, S., & Matsuo, H. (2021). The impact of COVID-19 on university students’ academic motivation, social connection, and psychological well-being. Scholarship of Teaching and Learning in Psychology. https://doi.org/10.1037/stl0000294).

The above issue should be also included in limitations.

 Discuss the relationship between motivation and employment status.

 While the strategies for overcome the issues in business students are well argued, the healthcare students’ counterpart is poor and needs a deepening.

Please see the following papers to improve this section:

 Tahara, M.; Mashizume, Y.; Takahashi, K. Mental Health Crisis and Stress Coping among Healthcare College Students Momentarily Displaced from Their Campus Community Because of COVID-19 Restrictions in Japan. Int. J. Environ. Res. Public Health 2021, 18, 7245. https://doi.org/10.3390/ijerph18147245

 Baker, S.R. Intrinsic, extrinsic, and amotivational orientations: Their role in university adjustment, stress, well-being, and subsequent academic performance. Curr. Psychol. 2004, 23, 189–202.

 Orsini, C.; Binnie, V.I.; Wilson, S.L. Determinants and outcomes of motivation in health professions education: A systematic review based on self-determination theory. J. Educ. Eval. Health Prof. 2016, 13, 19

Yun, M.R.; Lim, E.J.; Yu, B.; Choi, S. Effects of Academic Motivation on Clinical Practice-Related Post-Traumatic Growth among Nursing Students in South Korea: Mediating Effect of Resilience. Int. J. Environ. Res. Public Health 2020, 17, 4901. https://doi.org/10.3390/ijerph17134901

 Conclusion

The conclusion might be written in a way that encompasses all mentioned issues.

Finally, I would like to congratulate the authors because they faced a challenge, developing complex research in comparing two profoundly different groups. This is the additional point, which makes this manuscript original, in comparison to published literature. Surely the study is also useful for the governance of universities that deal with a plurality of courses and must respond to diverse needs. I am sure that, with the appropriate adjustments, the work will have a good scientific impact. I would accept this manuscript if the comments will be addressed properly.

Author Response

Response Letter

Manuscript ID: healthcare-1858411

"Comparison of academic motivation between business and healthcare students in online learning: A concurrent nested mixed-method study”

Dear Reviewers,

Thank you for your helpful feedback. We have systematically revised our manuscript addressing the points you have raised. Please see our responses below. We hope this revised paper is now acceptable for publication. We extend our sincere gratitude to you for your feedback that has significantly helped to strengthen the paper.

Reviewer 2

Reviewer 2’s comment 1

The aim of this manuscript is to evaluate the academic motivation of online students, focusing on business and healthcare students. It employed a concurrent nested mixed-method design, that gave priority to quantitative data, while qualitative findings were used to gain a better understanding of the students’ motivation (intrinsic/extrinsic). The quantitative analysis was performed by the Academic Motivation Scale, the qualitative module included seven items. The results showed that high intrinsic motivation was associated with healthcare students, while high extrinsic motivation was associated with business students.

The research design is appropriate, and quantitative and qualitative aspects of the study are clearly linked.

As regards the topic, this manuscript touches a significant area, and may represent a considerable contribute to the ongoing educational research since there is few evidence on this topic.

 Even if the manuscript provides an organic outline, I have some points to mention, which may help to improve its quality. For these reasons, the manuscript requires major changes.

Please find below a list of comments on my review of the manuscript:

Introduction

Major points

LINES 32-49

As the main object of the study includes health and business students, rather than a general analysis of online teaching advantages and disadvantages, a more detailed explanation of challenges in these specific groups should be provided.

Business schools are in a particularly advantageous position to adopt online teaching because business studies do not require as much practical skills as medical studies (see Szopiński T, Bachnik K. Student evaluation of online learning during the COVID-19 pandemic. Technol Forecast Soc Change. 2022;174:121203. doi:10.1016/j.techfore.2021.121203).

In healthcare schools, online teaching is suitable for pre-clinical courses (see Bianchi S., Gatto R., Fabiani L. Effects of the SARS-CoV-2 pandemic on medical education in Italy: Considerations and tips. EuroMediterranean Biomed. J. 2020;15:100–101. doi: 10.3269/1970-5492.2020.15.24), however, clinical courses were negatively affected by the interruption of internships (see Vijayan, R. Teaching and Learning during the COVID-19 Pandemic: A Topic Modeling Study. Educ. Sci. 2021, 11, 347. https://doi.org/10.3390/educsci11070347).

Expanding this section would ensure a fair balance between the advantages and disadvantages of the two groups, leading the reader from a general subject area to a particular field.

Authors’ response 2-1

Thank you for your helpful and thoughtful feedback, and suggestions. In line with your comment, challenges that these students face in terms of motivation and online education contexts during COVID-19 are added using the sources suggested.

Reviewer 2’s comment 2

LINE 67 AND FOLLOWING

A reference about the role of intrinsic/extrinsic motivation on academic performance in online settings is required (see Meng, X. and Hu, Z. 2022, "The relationship between student motivation and academic performance: the mediating role of online learning behavior", Quality Assurance in Education. https://doi.org/10.1108/QAE-02-2022-0046)

Authors’ response 2-2

In line with your comment and suggestion, now the insights from the suggested paper are added.

Reviewer 2’s comment 3

Minor points

Line 61 and following

Abbreviations should be defined the first time they appear in the main text and should be added in parentheses after the written-out form (intrinsic/extrinsic motivation). Please use abbreviations throughout the text.

Authors’ response 2-3

We are sorry for this error. It is now amended.

Reviewer 2’s comment 4

Materials and Methods

Major points

Please add a table with demographic data of the participants including the employment status

Authors’ response 2-4

In line with your comment, a table about demographic data is added.

Reviewer 2’s comment 5

Minor points

Line 183 the paragraph refers only to the quantitative analysis so, to avoid confusing the reader, please delete the sentence “and a qualitative component comprising seven questions.”

Authors’ response 2-5

In line with your comment, it is deleted.

Reviewer 2’s comment 6

Line 204

I think there is a mistake in the heading, probably you meant “qualitative”

Authors’ response 2-6

Thank you for spotting. Now it is amended.

Reviewer 2’s comment 7

Results

Major points

The employment situation affects students' motivation, so the following concepts should be properly in included in materials and methods (in the table of demographic data) and addressed and clarified in the discussion section:

Lines 277-278 “business students were often enrolled in the course solely on the request of their employers”

Lines 327-330 “The majority of online healthcare students at our institution are registered professionals, and many of them have other non-academic commitments (e.g., childcaring). Their life context is often more restricted than typical young students who have just graduated from high school to afford education.”

Authors’ response 2-7

Thank you for your insights. Clarification has been added to these sentences that what we referred to were those students who were in employment. Unfortunately we did not collect data about the employment status of participating students; this is now noted in the limitation.

Reviewer 2’s comment 8

Discussion

Major points

Please provide a summary of both qualitative and quantitative results at the beginning of the section and then start the discussion in a comprehensive way. The separation of the two section weighs down the reading and generates repetitions of sentences and concepts.

Since the study was conducted during pandemic a brief analysis of the pandemic impact on students' academic motivation, is required (See Marler, E. K., Bruce, M. J., Abaoud, A., Henrichsen, C., Suksatan, W., Homvisetvongsa, S., & Matsuo, H. (2021). The impact of COVID-19 on university students’ academic motivation, social connection, and psychological well-being. Scholarship of Teaching and Learning in Psychology. https://doi.org/10.1037/stl0000294).

The above issue should be also included in limitations.

Authors’ response 2-8

Thank you for your suggestion. A summary of our results is placed at the beginning of the section. The two sets of results are now discussed in a more comprehensive way. Moreover, using the suggested source, reference to the pandemic is added in the discussion and the limitation.

Reviewer 2’s comment 9

Discuss the relationship between motivation and employment status.

Authors’ response 2-9

In line with your comment, this relationship is discussed in the limitation.

Reviewer 2’s comment 10

While the strategies for overcome the issues in business students are well argued, the healthcare students’ counterpart is poor and needs a deepening.

Please see the following papers to improve this section:

 Tahara, M.; Mashizume, Y.; Takahashi, K. Mental Health Crisis and Stress Coping among Healthcare College Students Momentarily Displaced from Their Campus Community Because of COVID-19 Restrictions in Japan. Int. J. Environ. Res. Public Health 2021, 18, 7245. https://doi.org/10.3390/ijerph18147245

 Baker, S.R. Intrinsic, extrinsic, and amotivational orientations: Their role in university adjustment, stress, well-being, and subsequent academic performance. Curr. Psychol. 2004, 23, 189–202.

 Orsini, C.; Binnie, V.I.; Wilson, S.L. Determinants and outcomes of motivation in health professions education: A systematic review based on self-determination theory. J. Educ. Eval. Health Prof. 2016, 13, 19

Yun, M.R.; Lim, E.J.; Yu, B.; Choi, S. Effects of Academic Motivation on Clinical Practice-Related Post-Traumatic Growth among Nursing Students in South Korea: Mediating Effect of Resilience. Int. J. Environ. Res. Public Health 2020, 17, 4901. https://doi.org/10.3390/ijerph17134901

Authors’ response 2-10

In line with your comment, discussion for the results from healthcare students is now strengthened by referring to the suggested sources. Thank you.

Reviewer 2’s comment 11

Conclusion

The conclusion might be written in a way that encompasses all mentioned issues.

Finally, I would like to congratulate the authors because they faced a challenge, developing complex research in comparing two profoundly different groups. This is the additional point, which makes this manuscript original, in comparison to published literature. Surely the study is also useful for the governance of universities that deal with a plurality of courses and must respond to diverse needs. I am sure that, with the appropriate adjustments, the work will have a good scientific impact. I would accept this manuscript if the comments will be addressed properly.

Authors’ response 2-11

In line with your comment, now the conclusion is amended accordingly. Thank you so much for kind words!

Round 2

Reviewer 1 Report

Thanks for the revisions. The manuscript looks quite good at present.

Reviewer 2 Report

The authors addressed all the issues, well done!

Please check the reference list, there are some mistakes (for example ref n. 21, 23 and 24)